# Changes in Quality of Life in Elementary School Children—The Health Oriented Pedagogical Project (HOPP)

**DOI:** 10.3390/sports7010011

**Published:** 2019-01-03

**Authors:** Per Morten Fredriksen, Helene Kristine Olsen, Trine Johansen Meza

**Affiliations:** Department of Health Sciences, Kristiania University College, Oslo 0152, Norway; permorten.fredriksen@kristiania.no (P.M.F.); helene.k.olsen@hotmail.com (H.K.O.)

**Keywords:** Quality of Life, ILC, elementary school children, parents

## Abstract

Background: Quality of life (QoL) studies may provide information of change in health status in the population. Few studies have followed up previous cross-sectional studies to investigate any change in the QoL status of healthy children. The aim of the current study is to compare QoL in children 6–12 years old in two large cross-sectional studies of healthy children completed a decade apart. Methods: In the current study children and parents from nine elementary schools (n = 2816) were included in a cross-sectional study investigating children’s QoL. Using the Life Quality in Children and Adolescents (ILC), completed by both children and parents, the global QoL-score was estimated for 2297 children and 1639 parental reports. These results were compared to a similar study performed in 2004. The scores from both studies were divided into categories of below average, average and above average QoL. The percentage change of QoL between the two studies is presented. Results: Our results show that parents report that more children have below and above average QoL in 2015 compared to 2004. In contrast, more children scored in the average and above average QoL category in 2015 than in 2004. Conclusion: Parents reported lower QoL and children higher QoL in 2015 compared to 2004.

## 1. Introduction

Children is a vulnerable group dependent on adults for their opportunities to a good quality of life (QoL) [1]. To reveal changes in a child population over time, epidemiological studies on a healthy children population’s QoL are necessary [1]. Such knowledge is important for parents, school counsellors, child- and family therapists, as well as for the health authorities and politicians [2]. A negative trend may call for measures and some cross-sectional studies do show declining QoL with increasing age in adolescents [3,4]. Several studies have been conducted on adults, however fewer studies have been performed for children and adolescents [1].

To implement studies on young children, the QoL instruments have to be of a generic character to encompass the various domains which comprise QoL. A suitable QoL instrument should include negative and positive states of physical, mental, and social domains, and adding questions of family, school, autonomy, leisure time activity, and the child’s environment [5,6,7,8,9,10,11,12,13]. Instruments for children must be short, simple, and easy to complete, taking into account cognitive developmental level in reading skills and emotional maturity to match different developmental stages [10,11]. Despite that in recent years numerous instruments have been developed for estimating QoL in children [5,6,7,8,9,10,11,12,13,14,15], few instruments designed for children manage to incorporate these domains [16,17]. 

Using an instrument measuring generic QoL as a screening tool does not necessarily predict specific medical psychiatric conditions, but can rather give an indication of a population’s general well-being. According to Cummins, QoL appears to be a stable trait, as negative events tends to be short-lived. Cummins suggests that there is a system that homeostatically maintains subjective QoL, keeping it within certain predefined limits [16]. A mapping of QoL in different populations across decades may therefore not reveal change at an individual level, but rather on a group level. As society changes, children’s QoL should be followed to uncover variations, as secular changes in QoL may call for regular follow-ups in culturally similar populations with identical instruments. 

The primary aim of the current study was to assess the level of QoL in children in the 6–12 old year age group in 2015, and compare the results with the Norwegian reference material from 2004 to detect any changing trends in QoL. The hypothesis was to detect any difference in response using the ILC in two cross-sectional studies, a decade apart.

## 2. Methods

### 2.1. Population and Sample

As part of the Health Oriented Pedagogical Project (HOPP), parents and children in nine elementary schools received an invitation to participate in the project [18]. Seven schools were located in Horten municipality, 100 km south of Oslo, Norway, and two schools were located in the Oslo metropolitan area (Lørenskog and Bærum municipalities) [18]. Both areas are predominantly upper middle class areas. Informed consent was received from the parents. The data collection was performed in 2015. Of a population of 2816 children, 2297 children (82%) were approved to participate through their parents and received a quality of life questionnaire (Figure 1). In total, 2140 (93% of 2297) and 1639 (71% of 2297) responses of children and parents were completed. Figure 2 shows an evenly distribution of children across age-groups in the 2015 data. The results from the study in 2004 included data from 1987 (71%) children and 1777 (89%) parents [19]. The sample was collected from Sør-Trøndelag county, in the middle part of Norway. There are geographical differences between the samples, however—Norway is a very homogenic society—hence small differences between the samples are expected.

### 2.2. Quality of Life

A Norwegian version of the generic 7-item Inventory of Life Quality in Children and Adolescents (ILC) was used to assess various QoL items, recalling the past week [20]. The ILC includes a global QoL-score using seven items, concerning school performance, family relations, peer relations, autonomy in play, physical health, mental health, and a global assessment of wellbeing. The parental version corresponds to the children’s version [21]. 

The questionnaire have pictograms for the youngest children making it easy to complete for all age groups [19,20]. ILC consist of both a children’s and a parental report, and from 2004, a Norwegian reference material is available [19]. 

According to the manual each item was rated on a 1–5 Likert scale, with 1 presenting no problems and 5 indicating a severe problem. In addition to scoring the raw data, two types of scores was calculated: problem score (PR_0–7_) and quality of life score (LQ_0–100%_). The PR_0–7_ is computed by dichotomizing each of the seven items, such that ratings of 1 or 2 = 0 (no problems) and ratings of 3, 4, or 5 = 1 (present problem). A high score in PR_0–7_, indicates more problems. Based on the PR_0–7_ of each item, a total problem score is calculated by summarizing each score. This give a problem score range from 7–35, with maximal scores indicating more problems. LQ_0–100%_ is calculated when subtracting 35 (the highest problem score) by 7 (the lowest problem score) in the total problem score (PB_35_), which equals 28 (LQ_0–28_). LQ_0–28_ is then calculated by multiplying the mean of the seven items by seven, and the LQ_0–100%_ is the LQ_0–28_ divided by 28 and multiplied by 100. Hence, high values in LQ_0–100%_ indicate higher QoL [19]. As an example; if a child scores 11 on the total problem score (7–35), 11 is subtracted from 35, which equals 24 on the LQ_0–28_ score. The number 24 may be used as a result in LQ_0–28_-score. To convert it to percentage 24 is divided by 28 and multiplied with 100 giving 85.7 as a LQ_0–100%_-score. A percentage of 85.7 indicate a high QoL.

### 2.3. Procedure

For parents, a digitalized parental version of the ILC was distributed using e-mail, or alternatively distributed by satchel post for those parents where e-mail addresses were missing. The informed consent and questionnaires were attached. In addition to age, gender and the informant’s relation (mother/father/other) to the child, parents educational level (primary school, high school, bachelor degree, and master degree or higher), was collected. To keep the anonymity, an ID-code was included in the mail, one code for each child for families with multiple children. Non-responding parents received a reminder after one week, per e-mail, alternatively by satchel post. 

The children’s report was completed on a hard copy during school hours under supervision for the youngest pupils. The test supervisors were trained in giving instructions for each question according to the ILC-manual. Fourth to sixth grade pupils managed to complete the form without assistance. Any pupil in need of assistance for example with language, could get this from the test supervisors.

### 2.4. Comparison between 2015 and 2004

A Norwegian manual for ILC contains a reference material was collected in 2004 and used in the manual for the Norwegian ILC [19]. These data was utilized for comparison with data collected in the current study. The ILC-manual give the scoring categories in percentiles. Three categories were used; (1) Children having below average QoL (below average), (2) Children having average QoL (average), and (3) Children having above average QoL (above average).

To quantify the percentage of children who scored in the three categories in the present study the cumulative percentage was used. The reference data display the distribution of LQ_0–28_, and classifies the results for each age and gender category. In the present study, the LQ_0–28_ is converted into LQ_0–100%_ prior to presenting the results. To find the similar category in the present material as in the manual the corresponding cut-off point for “below average”, “average” and “above average” was used in the HOPP-study. 

As an example: If the highest value in the manual for “below average” was 17 in the LQ_0–28_ scale for 6–8-year-old boys, the corresponding cumulative percentage was registered in the manual (e.g., 12.3%) and concurrently in a frequency distribution for the results in the HOPP-study (e.g., 18.4%). This indicates a 6.1% increase of the number of participants in the “below average” category in the HOPP-study compared to the 2004-manual, which imply a decrease in QoL for an increased number of children.

The cumulative percentage was based on the following: (1) the highest cumulative percentage for the category “below average” was used as the representative value of this category, (2) the “average” cumulative percentage is the range between highest value of “average” and highest value of the “below average” and (3) the “above average” − “average” indicate the percentage of “above average”. The latter meant subtracting 100% with the highest “average” percentage, e.g., 100% − 83.5% = 16.5%, which show that 16.5% of the reference material belongs in the “above average” category. 

To find the actual change in percentage the next step involves dividing the acquired percentage from the different categories in the HOPP-study, based on gender and age group, with the corresponding category in the manual [19]. Positive values indicate an increase in the number of children in the different categories, negative values a decrease. In addition, by subtracting the results from the reference material with the results in present study, then divide the current value with the reference material, the variation of the difference was found.

Depending on the category in question the interpretation may give a positive or negative result. It is the difference in number of children in each category, measured as percentage change in each category, that is the crucial argument. That is, if more children are categorized as “below average”, the QoL in 2015 are worse than in 2004. If a higher percentage of children were categorized in the “below average” category it was considered negative. If a higher percentage of children were found in the “above average” category, it was considered positive. An increase in the “average” category at the expense of a reduction in the “below average” was also considered positive. The raw scores are given as mean values with ±1 SD, as are the sum of problem score PR_0–7_ and the QoL scores of LQ_0–100%_. A *p*-value of 0.05 was considered significant. 

### 2.5. Statistics

To identify changes in the categories of QoL (“below average”, “average”, “above average”) between the HOPP-study in 2015 and the 2004 reference material, Chi^2^ analyses were utilized using GraphPad QuickCalcs (Graphpad.com). To calculate if observed changes of QoL between the reference material of 2004 and data collected in this study 2015 were statistically significant, a t-test for percentage in Winks SDA was utilized (Texasoft.com). 

### 2.6. Ethical Consideration

Informed consent was acquired from all parents prior to inclusion in the study. The Regional Committee for Medical and Health Research Unit approved the study (reference number 2014/2064/REK sør-øst).

## 3. Results

In total, 1080 (51%) boys and 1060 (49%) girls completed the questionnaire (Figure 1). The missing 162 children were, due to various reasons as illness, dentist visits, physician visits, travels, or exams, absent on test day. The corresponding response for parents were 826 (51% of 1639) boys and 806 (49% of 1639) girls (Figure 1). As some informants (n = 116) completed the questionnaire without stating their relation (mother/father/other) to the child, 393 fathers (26% of 1516), 1114 mothers (74%), and 9 (1%) guardians were registered. 

To identify any change of QoL in the child population between 2004 and 2015 the reference data from the ILC manual of children (n = 1987) and parents (n = 2563) in 2004 are used [19]. The ILC manual divide the results into below average, average and above average for both children and parental response. 

In the present study, 286 (17% of 1639) parents score their children’s QoL below average. The majority, n = 912 (56% of 1639), rate their children’s QoL at average, and 441 (27% of 1639) scored above average (Table 1). 

Due to the lack of reference data for 6 and 7 years old children, [19], only data for 8–10 years old and 11–12 years old are shown in Table 2, hence the total number of children are n = 1421. Analysis of the data collected from the children’s response, shows that 123 (9%) out of 1421 children consider their QoL to be below average (Table 2). 1030 (73%) children scored average QoL, and 268 (19%) children revealed QoL above average. 

Compared to the findings in 2004, an increased number of parents consider their children to have lower QoL in the present study (Figure 3) [19]. This is evident for boys in the age groups 8–10 and 11–12 and in contrast to the 6–8-year-old group where no significant increase was found (Figure 3). In the 8–10-year-old group a 10% increase (*p* < 0.01) in the number of parents reporting the children to have QoL below average is found (Figure 3). In the 11–12-year-old group, more than 15% (*p* < 0.0001) increase in parents scoring their children’s QoL below average was found. Many parents reported lower QoL in girls in 2015 compared to 2004 as well, however, the numbers are not significant (Figure 3). 

In contrast with the findings from the parents, the findings from the children show the opposite tendency as there currently are fewer children in the “below average” category compared to 2004. This tendency is significant for 11–12-year-old boys (*p* < 0.001) and girls (*p* < 0.0001), however not for the 8–10-year-old. As no data were available for the youngest age group in the reference data, comparison was not performed. Interestingly, twice as many parents consider their children’s QoL to be less than average than do the children themselves. 

When comparing the reference data with HOPP-data, fewer boys and girls, were scored in the category “average” by their parents in 2015, as shown in Figure 4. Especially for boys 11–12-year-old a significant reduction (*p* < 0.0001) was shown, with a decrease of almost 30%. A decrease of 10% (*p* < 0.001) was displayed in boys in 8–10-year-old boys. Parents of girls also demonstrate a drop, however, less pronounced (*p* = ns) than for the boys in all age categories. 

As shown in Figure 4, an opposite tendency is evident from the children’s responses with a significant increase (*p* < 0.01) for average QoL in the number of boys in the 8–10-year-old group. The oldest age group showed no significant increase. Only small, non-significant differences were observed for girls.

As shown in Figure 5, 13% (*p* < 0.01) boys in the 11–12-year-old group were categorized “above average” by the parents in 2015 compared to 2004. This was not displayed for girls in the same age group (Figure 5). Minor, non-significant, changes were disclosed in the two youngest age groups for boys. A non-significant increase in QoL was assessed for 6–8 years old girls in the “above average” category (Figure 5). 

When analyzing the scores given by the children a different pattern was displayed (Figure 5). A tendency towards fewer 8–10-year-old boys (*p* = ns) categorize themselves as being “above average” in 2015, as opposed to boys in the 11–12-year-old group. Significantly (*p* < 0.01) more 11–12-year-old girls report QoL in the “above average” category in the present study compared to 2004 (Figure 5).

## 4. Discussion

There are important differences between testing for effect of an intervention, to assess diagnoses and screening for QoL [21]. Comparing the data from the present study with the data from the ILC manual from 2004, our results indicate a polarization amongst the parent’s estimation of their children’s QoL, especially evident for boys. Compared to 2004, a percentage increase of parents score boys to have “below average” and “above average” QoL, especially for the oldest age group. Similar changes have occurred for girls, however, for the youngest age group. Fewer fluctuations were disclosed in the 8–10 and 11–12-year-old girls. According to Norman et al., a weakness in any QoL questionnaire is that small changes in response may be a direct consequence of human inability to discriminate within each item [22]. 

ILC has proven to be used both as a clinical tool for patients as well as a screening tool in research projects to estimate the QoL in a general population [19,21]. However, it is questionable if ILC may be used as a screening tool to reveal pathologically reduced QoL, requiring follow-up in child populations. The cut-off point for being in the “below average” category in the current data or reference material, is not by any means a cut-off point for pathology. Inferior QoL or wellbeing has to be paired with more severe pathology to signify intervention [19]. Nevertheless, the ILC may be used as an appropriate tool when assessing patient’s QoL, however, it may be less useful in detecting pathology on an individual basis in population screening scenarios [19,23]. A floor and ceiling effect may be one explanation, as a high ceiling effect causes low sensibility for change. This deflates the numerical gains for participants with very severe or minor disabilities, as the majority of participants have moderate disabilities. Mattejat et al. evaluated the floor (7%) and ceiling (14%) effect for ILC, and found both to be satisfactory across a 6-month period [24].

### 4.1. Differences Across Time

According to Wallander and Koot, few large studies on children’s QoL are completed, and even fewer have compared two large samples from the same country a decade apart in order to disclose any change in children’s QoL [1]. Such data could give valuable information to health authorities, school counsellors, child therapists, family therapists and politicians as negative trends may induce appropriate adjustments. Performing such comparisons would be feasible if there are culturally, socioeconomically and socially comparable populations. The current dataset is obtained in the south-east part of Norway, whilst the ILC-manual is obtained further north. This may cause some cultural and socioeconomic differences. Unfortunately, data from the ILC-manual does not include any information about socioeconomic status which could have explained differences. Despite this, relatively objective comparison is expected as the Norwegian society is quite homogenous with a well-functioning welfare system and small socioeconomic differences. In addition, the sample size, the use of parental and children’s report and similar age and gender distribution imply comparable results. Given the similarity in the population from 2004 and 2015, the difference in QoL in the two populations indicates a need for regular follow-up of elementary school children in order to detect any negative or positive trends. 

In the present study parents are more concerned for their children than a decade ago, as they categorize their children in the “below average” category. However, parents also classify their children as having QoL “above average”; hence a polarization is evident. The latter may be explained by the increase in higher education in the Norwegian population. This is underlined by data from Statistics Norway that show that approximately 23% of the adult population had higher education (3 years or higher after high school) in 2004, compared to 31% in 2014 (latest update) [25]. The increase in the level of higher education is equally distributed in the counties the studies are collected from [25].

In this study, the parental reply disclosed more children having QoL “below average” compared to 2004. Data from Statistics Norway from 2000 to 2010 reveal an increase in the time adolescents spent with computer games daily (2000—14%, 2010—20%) and internet activity (2000—19%, 2010—58%) [26]. More potentially social activities such as TV-watching (2000—77%, 2010—73%), social interaction (2000—82%, 2010—64%), and cultural participation (2000—20%, 2010—9%) was reduced [26]. Data from children (9–12 years of age) display a similar trend in percentages for computer games (40%) and internet activity (33%). TV-watching (84%) is still higher for children, but social interaction (69%) and cultural participation (7%) are at the same level as for the adolescents. This being highly speculative of course, but the increased “below average” results may partly and theoretically be explained by parents being more worried for their children’s lack of social interaction, perhaps in combination with more solitude due to increased time spent in front of different screens.

### 4.2. Parental and Child Response

Several studies have shown that parents and children diverge widely in their responses [27,28]. The difference has partly been explained by the fact that the parents own emotional involvement colors their responses, and that mothers emphasize children’s disease more than fathers [29]. Suggestively, parents rate their children as having lower total wellbeing than what the children do. Jozefiak state that QoL should primarily be attained from the child if possible, and that parental reports may represent a supplement [28,30]. Data from the present study reveal a discrepancy between parents and children as to how the distribution of QoL are among the three categories. Fewer children (5–10%) classify themselves to have QoL “below average”, as compared to the ILC manual from 2004. It appears to be at the expense of an increased number of children having “average” or “above average” QoL. The boys in the youngest age group appear to have moved from QoL “above average” to an “average” QoL. This is opposed to the 11–12-year-old group of boys who consider themselves to have QoL on “average” and “above average” at the expense of QoL “below average”. Interestingly, the present study shows that the QoL in girls, both the 8–10 years and 11–12 years age-group, have transformed completely from the “below average” to “above average” category in the last decade. The reason for this is challenging to explain.

### 4.3. Limitations of the Study

A weakness of the comparison may be the difference in response rate, as the current study display an inferior response rate compared to ILC manual, especially for the parental response [19]. Approximately 30% of the parents did not respond in the current study, compared to only 10% in the reference material. It is not clear to what extent and direction this influence the outcome.

The results displayed in the ILC manual is based on cumulative percentage to reveal the number of children in each category. This may hinder the possibility to establish central tendencies as well as dispersion using mean and standard deviation. Again, this obstruct the possibility to create confidence intervals for the results of 2004. Despite this the use of cumulative percent do give a fairly good description of the dispersion of the data.

## 5. Conclusions

Overall, children in 2015 consider themselves to have higher QoL compared to children in 2004. The parents on the other hand reported a polarization 2015 as an increased number of children were categorized in “below” and “above average” QoL categories compared to 2004. The results from these two cross-sectional studies indicate that both parent’s and children’s perception of children’s QoL have changed the last decade. This underline the importance of regular follow-ups to reveal changes of QoL in the child population. These results may be used for politicians and other decision makers as guidelines as to what extent actions are needed. Future studies should emphasize to use similar method and sample size, and preferably supplement the data with parental socioeconomically status.

## Figures and Tables

**Figure 1 sports-07-00011-f001:**
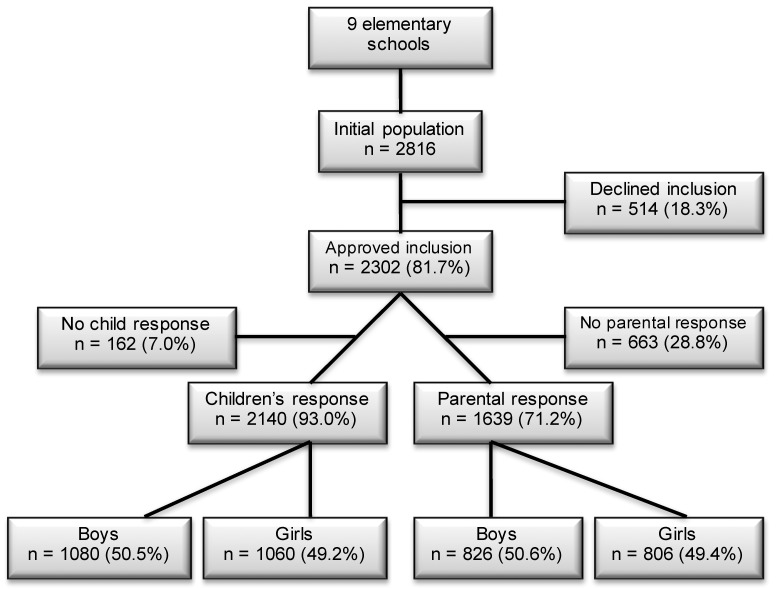
The flowchart displays the children and parents who responded on the Inventory of Life Quality of Children and Adolescents (ILC) in the Health Oriented Pedagogical Project (HOPP) in 2015. The children lost to follow-up were mainly due to illness at test day, dentist visits, physician visits, travels, or exams.

**Figure 2 sports-07-00011-f002:**
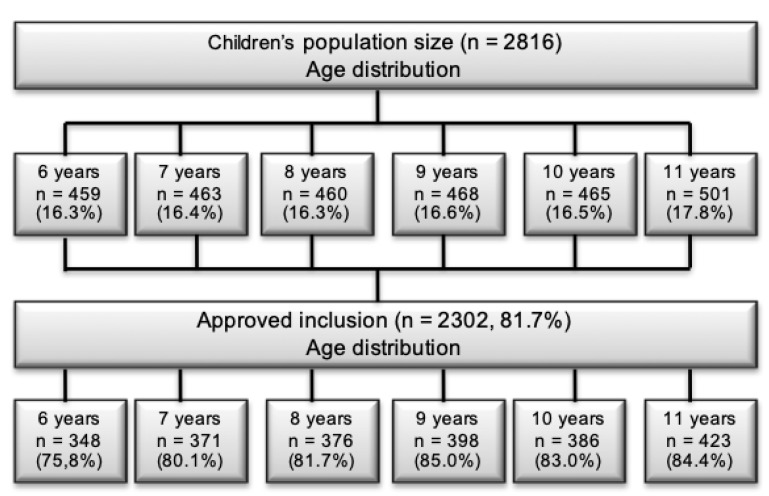
The figure displays the age distribution of the total population and of the sample that approved inclusion in the Health Oriented Pedagogical Project (HOPP) in 2015.

**Figure 3 sports-07-00011-f003:**
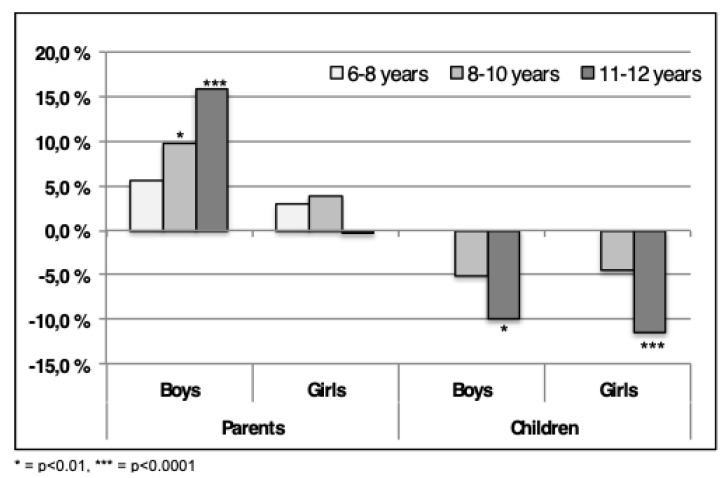
The differences in the percentage of children in the category “below average” between 2004–2015 are shown. Positive numbers indicate more children scores” below average” in 2015, negative indicate fewer children. The scores are both from parents and children using ILC.

**Figure 4 sports-07-00011-f004:**
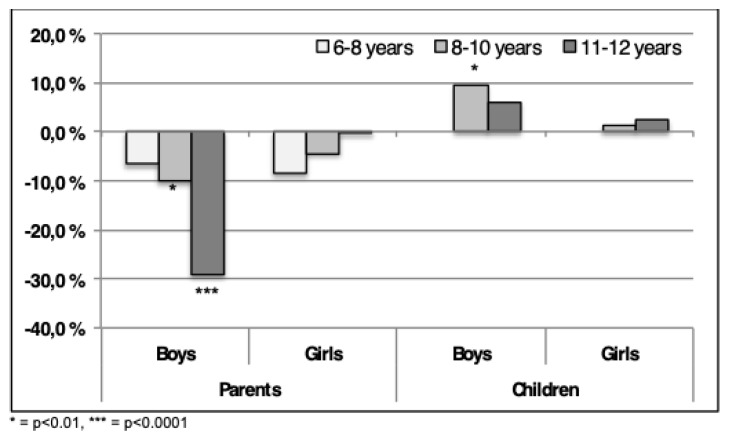
The differences in the percentage of children in the category “average” between 2004–2015 are shown. Positive numbers indicate more children scores “average” in 2015, negative indicate fewer children. The scores are both from parents and children, using ILC.

**Figure 5 sports-07-00011-f005:**
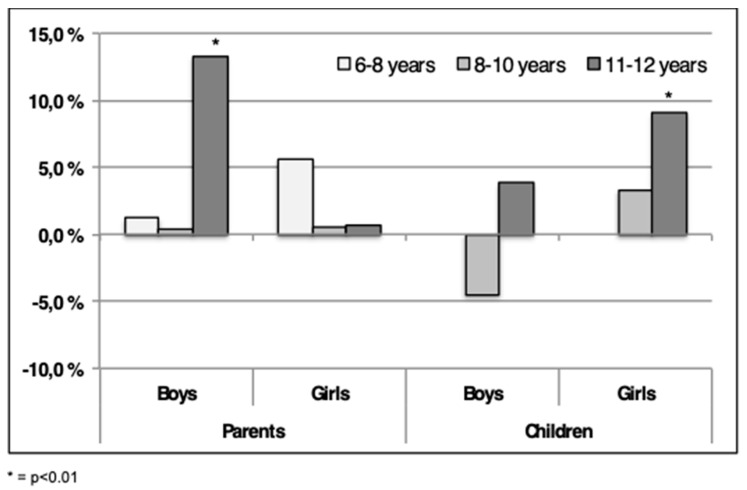
The differences in the percentage of children in the “above average” category between 2004–2015 are shown. Positive numbers indicate more children scores “above average” in 2015, negative indicate fewer children. The scores are both from parents and children using ILC.

**Table 1 sports-07-00011-t001:** The number of parents who score their children’s quality of life (QoL) “below average”, at “average”, and “above average” are displayed. The categories are retrieved from the Norwegian manual for ILC [19].

Age	Below Average		Average		Above Average	
	Boys	Girls	Boys	Girls	Boys	Girls
6–8 years	74	69	265	229	67	82
9–10 years	6	41	119	139	85	95
11–12 years	22	14	86	74	49	63
Total	162	124	470	442	201	240

**Table 2 sports-07-00011-t002:** The table displays the number children in each age category who scores their QoL “below average”, at “average”, and “above average”. The categories are retrieved from the Norwegian manual for ILC [19]. The 6- and 7-year-old age group are excluded due to the lack of data in the manual.

Age	Below Average		Average		Above Average	
	Boys	Girls	Boys	Girls	Boys	Girls
8–10 years	42	58	410	339	89	87
11–12 years	6	17	142	139	41	51
Total	48	75	552	478	130	138

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
