# Peer review of "Changes in Quality of Life in Elementary School Children—The Health Oriented Pedagogical Project (HOPP)"

_sports, 2019, doi:10.3390/sports7010011_

Round 1

Reviewer 1 Report

Dear Authors,

First of all I would like to send congratulations to you for interesting article and stated that subject is very important as well it have to be presented wider and especially to people who organize our life and work. They have to remember that Families should find time to common active time spend together to build strong base for future life especially for kids. 

According to my reviewer duties. Methodology and kind of presentation is general right. 

My main request is to return to some details. 

References.: 

page 1 line 35 and 38 - you present .... (9-13)(5-9) .... and ..... (5-8) (12-15) - such presentation is for me not clear and confusing. In this part of txt you list different elements

"negative and positive states of physical, mental and social domains, and adding questions of family, 34 school, autonomy, leisure time activity and the child’s environment (9–13)(5–9)."  please try to each of this things put number it will be more clear to understand and find suitable references.
Similar situation is in next line 38
It must be corrected..  

 Please check also other places

At the end of Discussion sub-paragraph 4.2 - line 290 - 305 Please very detailed references number and Name because you showed e.g. Jozefiak number 31 - and it is possible to find the only Jozefiak 30 or 34?!
I did not found Belmann et al. (30)
I did not found references number 32, 33, 34
Please once or twice more times check every references and agreement with number and place of reference.

2. Graphs 

Your work contain too much additional marks "?" in many places. Probably during copy and paste the graph to document something went wrong. It is general technical suggestion to improve quality of presentation.

3. Edytor's details - Results page 5 line 195 in brackets you put space between number and % symbol e.g. 1080 (51 %) should be 1080 (51%) - check in other parts of document

Please meet my suggestions as friendly and support on the way to improve your style of article preparation. I wish you only positive things!  

Author Response

Reviewer 1:

Dear Authors,

First of all I would like to send congratulations to you for interesting article and stated that subject is very important as well it have to be presented wider and especially to people who organize our life and work. They have to remember that Families should find time to common active time spend together to build strong base for future life especially for kids. 

According to my reviewer duties. Methodology and kind of presentation is general right. 

My main request is to return to some details.

References.: 

page 1 line 35 and 38 - you present .... (9-13)(5-9) .... and ..... (5-8) (12-15) - such presentation is for me not clear and confusing. In this part of txt you list different elements

"negative and positive states of physical, mental and social domains, and adding questions of family, 34 school, autonomy, leisure time activity and the child’s environment (9–13)(5–9)."  please try to each of this things put number it will be more clear to understand and find suitable references.
Similar situation is in next line 38
It must be corrected.. 

Response: All references are now corrected.

Please check also other places

At the end of Discussion sub-paragraph 4.2 - line 290 - 305 Please very detailed references number and Name because you showed e.g. Jozefiak number 31 - and it is possible to find the only Jozefiak 30 or 34?! 
I did not found Belmann et al. (30)
I did not found references number 32, 33, 34
Please once or twice more times check every references and agreement with number and place of reference.

Response: All references are now corrected.

2. Graphs 

Your work contain too much additional marks "?" in many places. Probably during copy and paste the graph to document something went wrong. It is general technical suggestion to improve quality of presentation.

Response: This must have been a conversion problem from Word to PDF. We are unable to find these marks.

3. Edytor's details - Results page 5 line 195 in brackets you put space between number and % symbol e.g. 1080 (51 %) should be 1080 (51%) - check in other parts of document

Response: Corrected  

Reviewer 2 Report

The topic of the study is valuable to understand children and adolescent, as well as their parents’ perceptions of quality of life. The cross sectional study over a decade is commendable. However, the paper needs to be proof read before submission. There were times when I find it difficult to understand what the authors were conveying. The following are suggestions for the authors to consider:

Abstract

Please specify that both the parents and the children completed the survey.

Rephrase this sentence for clarity: “Parents reported more children to have below and above average QoL in 2015 compared to 2004.” This sentence seems confusing to someone who have not read the paper before the abstract.

Rephrase this sentence for clarity: “Parents reported lower QoL and children higher QoL in 2015 compared to 2004.”

Introduction

Line 26: Change “an” to “a” in this sentence: “….epidemiological studies on an healthy children…”

Line 36: Change “take” to “taking” in this sentence: “…simple and easy to complete, take into account cognitive developmental…”

Line 43: Rephrase this sentence for clarity: “Cummins suggest that people have a cognitive homeostatic control over their QoL….”

Methods

Figure 1: The “Approved inclusion n = 2302 (81.7%)” in figure 1 is not the same number as “2297 children (82%) approved to participate” (line 59). Please rectify the error.

Line 83: Please describe the reference material in this sentence: “…and from 2004, a Norwegian reference material is available.” Do you mean the manual that is mentioned in the next sentence? Please be consistent with wording.

Lines 91-95: This section is confusing: “LQ0-100% is calculated when subtracting 35 (the highest problem score) by 7 (the lowest problem score) in the total problem score (PB35), which equals 28 (LQ0-28). LQ0-28 is then calculated by multiplying the mean of the seven items by seven, and the LQ0-100% is the LQ0-28 divided by 28 and multiplied by 100. Hence, high values in LQ0-100% indicate higher QoL (19).” Perhaps giving an example of how LQ is calculated would be helpful.  

Line 107: Change “form” to “from” in this sentence: “Any pupil in need of assistance for example with language, could get this form the test supervisors.”

Line 131: Do you mean “subtracting” instead of “dividing” in this sentence: “The latter meant dividing 100% with the highest “average” percentage, e.g. 100% - 83,5 = 16.5%....”

Results

Lines 162-164: This sentence is unclear: “Some parents (n=116) completed the questionnaire without stating parenthood; hence 393 fathers (26% of 1516), 1114 mothers (74%) and 9 (1%) guardians were registered.” Please modify.

Line 188: Replace “a” with a comma in this sentence: “In the 11-12-year-old group a more than15% (p<0.0001)…”

Lines 203-204: This sentence does not fit in this paragraph which is interpreting figure 3 (below average data): “In contrast, eight percent more parents score their children to be “above average” in QoL than what was evident from the children’s score.”

Is it possible to further break down the components of QoL (i.e., school performance, family relations, peer relations, autonomy in play, physical health, mental health and a global assessment of wellbeing) to examine which components have changed over the decade? This could add valuable information that will have implications on the health of children and adolescent.

Discussion

Lines 260-261: This sentence is unclear: “Ideally this demands culturally, socioeconomically and socially comparable populations.” Please rephrase.

Author Response

Reviewer 2:

The topic of the study is valuable to understand children and adolescent, as well as their parents’ perceptions of quality of life. The cross sectional study over a decade is commendable. However, the paper needs to be proof read before submission. There were times when I find it difficult to understand what the authors were conveying. The following are suggestions for the authors to consider:

Abstract

Please specify that both the parents and the children completed the survey.

Response: Done.

Rephrase this sentence for clarity: “Parents reported more children to have below and above average QoL in 2015 compared to 2004.” This sentence seems confusing to someone who have not read the paper before the abstract.

Response: “Our results show that parents report that more children have below and above average QoL in 2015 compared to 2004”.

Rephrase this sentence for clarity: “Parents reported lower QoL and children higher QoL in 2015 compared to 2004.”

Response: “It is evident from the results that a higher percentage of the parents report that their children have lower QoL in 2015 than in 2004. In contrast, from the children’s responses we observe that they report higher QoL in 2015 compared to 2004.”

Introduction

Line 26: Change “an” to “a” in this sentence: “….epidemiological studies on an healthy children…”

Response: Done

Line 36: Change “take” to “taking” in this sentence: “…simple and easy to complete, take into account cognitive developmental…”

Response: Done

Line 43: Rephrase this sentence for clarity: “Cummins suggest that people have a cognitive homeostatic control over their QoL….”

Response: “Cummins suggest that there is a system that homeostatically maintains subjective QoL, keeping it within certain predefined limits

Methods

Figure 1: The “Approved inclusion n = 2302 (81.7%)” in figure 1 is not the same number as “2297 children (82%) approved to participate” (line 59). Please rectify the error.

Response:  The figure is corrected

Line 83: Please describe the reference material in this sentence: “…and from 2004, a Norwegian reference material is available.” Do you mean the manual that is mentioned in the next sentence? Please be consistent with wording.

Response: Changed the text to make it more clear.  

Lines 91-95: This section is confusing: “LQ0-100% is calculated when subtracting 35 (the highest problem score) by 7 (the lowest problem score) in the total problem score (PB35), which equals 28 (LQ0-28). LQ0-28 is then calculated by multiplying the mean of the seven items by seven, and the LQ0-100% is the LQ0-28 divided by 28 and multiplied by 100. Hence, high values in LQ0-100% indicate higher QoL (19).” Perhaps giving an example of how LQ is calculated would be helpful.  Se på

Response: “As an example; if a child scores 11 on the total problem score (7-35), 11 is subtracted from 35, which equals 24 on the LQ0-28 score. The number 24 may be used as a result in LQ0-28-score. To convert it to percentage 24 is divided by 28 and multiplied with 100 giving 85.7 as a LQ0-100%-score. A percentage of 85.7 indicate a high QoL.»

Line 107: Change “form” to “from” in this sentence: “Any pupil in need of assistance for example with language, could get this form the test supervisors.”

Response: Done

Line 131: Do you mean “subtracting” instead of “dividing” in this sentence: “The latter meant dividing 100% with the highest “average” percentage, e.g. 100% - 83,5 = 16.5%....”

Response: Yes, subtracting is correct.

Results

Lines 162-164: This sentence is unclear: “Some parents (n=116) completed the questionnaire without stating parenthood; hence 393 fathers (26% of 1516), 1114 mothers (74%) and 9 (1%) guardians were registered.” Please modify.

Response: “As some informants (n=116) completed the questionnaire without stating their relation (mother/father/other) to the child, 393 fathers ….”

Line 188: Replace “a” with a comma in this sentence: “In the 11-12-year-old group a more than15% (p<0.0001)…”

Response: Done

Lines 203-204: This sentence does not fit in this paragraph which is interpreting figure 3 (below average data): “In contrast, eight percent more parents score their children to be “above average” in QoL than what was evident from the children’s score.”

Response: The sentence is deleted.

Is it possible to further break down the components of QoL (i.e., school performance, family relations, peer relations, autonomy in play, physical health, mental health and a global assessment of wellbeing) to examine which components have changed over the decade? This could add valuable information that will have implications on the health of children and adolescent.

Response: This would be very difficult as the reference material is not easily assessable of rthese analyses.

Discussion

Lines 260-261: This sentence is unclear: “Ideally this demands culturally, socioeconomically and socially comparable populations.” Please rephrase. 

Response: “Performing such comparisons would be feasible if there are culturally, socioeconomically and socially comparable populations».

Reviewer 3 Report

Overall, interesting study.  I do have a  suggestions for your manuscript:

Attach a copy of the survey used for this study.

Ideas for further research, collect data from other countries beside Norway.

Author Response

Reviewer 3:

Overall, interesting study.  I do have a  suggestions for your manuscript:

Attach a copy of the survey used for this study.

Response: The ILC survey is not readily available as it is protected by sales and copying laws.

The project bought the survey to be used.

Ideas for further research, collect data from other countries beside Norway.

Response: This will be done in a study performed in collaboration with other countries in Europe.

Round 2

Reviewer 2 Report

The authors made the suggested changes to the paper.